A comprehensive prognostic and immunological analysis of hexokinase domain containing protein-1 (HKDC1) in pan-cancer

Liang Zhi
Zhang Tianhao
Huang Jiajia
Huang Zhixin
Zhao Zeyu
Cai Shirong caishr@mail.sysu.edu.cn
Ma Jinping majp@mail.sysu.edu.cn
Department of Gastrointestinal Surgery, The First Affiliated Hospital of Sun Yat-sen University , Guangzhou , Guangdong Province , China
Haraguchi Tokuko
Electronic publication date: 2025 Mar 19
Publication date: 2025
Volume: 13
Electronic Location ID: e19083
Received 2024 Oct 24; Accepted 2025 Feb 10
Copyright: ©2025 Liang et al.
Copyright year: 2025
Copyright holder: Liang et al.
License: This is an open access article distributed under the terms of the Creative Commons Attribution License, which permits unrestricted use, distribution, reproduction and adaptation in any medium and for any purpose provided that it is properly attributed. For attribution, the original author(s), title, publication source (PeerJ) and either DOI or URL of the article must be cited.
License URL: https://creativecommons.org/licenses/by/4.0/

Keywords: HKDC1, Pan-cancer, Prognosis, Immunological, DNA methylation

Funding: Guangdong Provincial Natural Science Foundation 2023A1515111152 Guangdong Provincial Health Commission A2024052 Guangzhou Nansha District Science and Technology Plan Project 2023MS003 Guangzhou Nansha District Science and Technology Plan Project 2024MS006 This study was supported by the Guangdong Provincial Natural Science Foundation project (No. 2023A1515111152); Guangdong Provincial Health Commission (No. A2024052); Guangzhou Nansha District Science and Technology Plan Project (No. 2023MS003); Guangzhou Nansha District Science and Technology Plan Project (No. 2024MS006). The funders had no role in study design, data collection and analysis, decision to publish, or preparation of the manuscript.

==============================
Background

Currently, research on the role of hexokinase domain-containing protein-1 (HKDC1) in neoplasm metabolism remains sparse. This study seeks to conduct a thorough investigation of HKDC1’s potential functions across thirty-three different tumor types, utilizing data obtained from The Cancer Genome Atlas (TCGA).

Method

We conducted a thorough data extraction from the TCGA database, subsequently employing R (version 4.2.2) and its associated software packages for detailed analysis. Our investigation centered on evaluating the differential expression and prognostic significance of HKDC1, while also examining its connections to tumor heterogeneity, mutation profiles, and RNA modifications. Furthermore, we analyzed the relationship between HKDC1 expression and tumor immunity utilizing the TIMER analysis approach.

Results

A comprehensive analysis of various tumor types has revealed that HKDC1 is significantly upregulated in many malignant tumors. Importantly, patients with elevated HKDC1 levels in their tumor tissues often experience poorer prognoses. The association between HKDC1 expression, immune cell infiltration, and the existence of immune checkpoints suggests a possible connection between the tumor microenvironment and HKDC1, alongside tumor advancement. Gene set enrichment analysis (GSEA) further substantiates the idea that HKDC1 may play a role in several critical pathways and biological processes associated with neoplasm. Additionally, the overexpression of HKDC1 is influenced by promoter methylation and alterations in DNA copy number amplification. Furthermore, in vitro experiments demonstrated that silencing HKDC1 resulted in a marked reduction in the proliferation, migration, and invasion capabilities of neoplasm cells.

Conclusion

Our initial pan-cancer analysis provided a comprehensive understanding of the oncogenic roles of HKDC1 across diverse cancer types. Moreover, HKDC1 has the potential to serve as a significant prognostic biomarker.

Introduction

In humans, tumor represents a major contributor to both illness and death, presenting a considerable obstacle to public health (Ginsburg et al., 2023). Currently, the conventional therapeutic strategies for neoplasm management encompass surgical excision, chemotherapy, and radiotherapy (GBD 2019 Viewpoint Collaborators, 2020). Despite these advancements, a subset of patients continues to exhibit unfavorable outcomes, including metastasis, recurrence, and suboptimal histological responses to initial therapeutic interventions (Swanton et al., 2020). Consequently, there exists a critical necessity to discover new biomarkers capable of precisely forecasting the pathological diagnosis and prognosis associated with tumor. Identifying these markers would enhance our understanding of tumor biology and facilitate the development of more effective and personalized therapeutic strategies.

In recent decades, the reprogramming of metabolism within the tumor microenvironment has become a hallmark of malignancy, underscoring the critical role that metabolic changes play in cancer progression. Emerging evidence underscores the potential of targeting cancer metabolism, particularly glucose uptake, as a therapeutic strategy for cancer treatment (Xiao et al., 2023). Cancerous cells display a unique metabolic characteristic referred to as the “Warburg effect.” This phenomenon is marked by a heightened dependence on glycolysis for energy production, even in the presence of sufficient oxygen (Yadav et al., 2024). Due to the Warburg effect, a large amount of lactic acid is produced, creating an acidic microenvironment for tumor cells, which promotes the invasion and migration of tumor cells (Hirschhaeuser, Sattler & Mueller-Klieser, 2011). Hexokinase 2 (HK2), an essential enzyme in glucose metabolism, plays a vital role in facilitating glucose uptake and utilization in cancer cells. Its increased expression has been observed in various tumor types (Fukushi et al., 2022). Other studies have shown that glycolysis has a unique advantage in enhancing drug resistance of tumors, that is, glycolytis-related enzymes play a role other than catalysis, including regulating transcription and inhibiting apoptosis, etc. For example, hexokinase is highly expressed in drug-resistant tumor cells and binds to VDAC embedded in the outer membrane of mitochondria to become mitochondrial binding hexokinase, thereby inhibiting apoptosis (Shoshan-Barmatz et al., 2015). Nonetheless, a relatively recent addition to the hexokinase family, known as hexokinase domain-containing protein 1 (HKDC1), has attracted interest owing to its enzymatic function as a hexokinase and its links to glucose metabolic processes (Irwin & Tan, 2014; Zapater et al., 2022). Studies have shown that intrahepatic HKDC1 plays a key role in glucose metabolism, insulin sensitivity and nutritional balance during pregnancy, and high expression of HKDC1 can improve overall glucose tolerance and enhance insulin sensitivity in liver and peripheral tissues (Khan et al., 2019). It has been reported in the literature that HKDC1 is expected to become a new drug therapeutic target for the treatment of lung cancer through a computation-based enrichment analysis of anticancer activity (Li & Huang, 2014). It has also been reported that HKDC1 is overexpressed in liver cancer tissues, and silencing HKDC1 can inhibit Wnt/β-catenin signaling and inhibit the proliferation and migration of liver cancer cells. High expression of HKDC1 is closely related to poor prognosis of liver cancer patients (Zhang et al., 2016).

Notwithstanding these compelling discoveries, the function of HKDC1 in the context of tumor biology is still inadequately examined, with a limited number of thorough studies assessing its association with different types of tumors (Ciscato et al., 2021; Khan et al., 2022). To fill this knowledge gap, we performed a comprehensive pan-cancer analysis using data from The Cancer Genome Atlas (TCGA). The primary objective of this study was to systematically evaluate the potential of HKDC1 as a biomarker across various cancer types. As illustrated in Fig. 1, our methodology encompassed a comprehensive design process and implementation strategies tailored to this research endeavor. Our findings demonstrate that HKDC1 holds significant prognostic relevance across a diverse array of tumors, underscoring its potential as a valuable biomarker for cancer prognosis and therapeutic strategies.

Figure 1 Analytical flowchart of this study.

Methods

Access to data and software

The dataset used in this research was sourced from the UCSC database (https://xenabrowser.net/datapages/), supplemented by corresponding normal tissues from the GTEx (Genotype-Tissue Expression) database. The analysis adhered to specific criteria, including a P-value threshold of 0.01, a log2 fold change (log2FC) cut-off of 1, and the requirement that TCGA (https://portal.gdc.cancer.gov/) normal data be aligned with GTEx data. All analyses were performed using R version 4.2.2.

Gene expression investigations

The gene expression profiles associated with pan-cancer were obtained from The Cancer Genome Atlas (TCGA), while profiles for normal tissues were sourced from the Genotype-Tissue Expression database, with both datasets accessed via the UCSC Xena Browser. Furthermore, the UCSC team facilitated the integration of datasets from both TCGA and GTEx, subsequently eliminating batch effects to ensure data consistency (Vivian et al., 2017).

Immune-related analysis

The TIMER database (Li et al., 2020) serves as a comprehensive online platform that rigorously investigates the interactions between neoplastic tissues and the immune system. This resource is designed specifically to assess the immune cell populations across 33 distinct cancer types. Additionally, in our study, the expression levels of 11 prominent immune checkpoint genes, including PDCD1, LAG3, BTLA, and others (Shibru et al., 2021), were also examined to conduct a correlational evaluation among these checkpoints.

Correlation examination in gene set enrichment analysis

To deepen our understanding of the specific biological role of the HKDC1 gene in tumorigenesis, we performed a Pearson correlation analysis to assess the relationship between HKDC1 expression levels and a variety of mRNAs derived from the transcriptome dataset of TCGA. Additionally, we conducted gene set enrichment analysis (GSEA) (Subramanian et al., 2005) using established gene sets from the Molecular Signatures Database version 5.0 by the R package ‘clusterProfiler’ (Yu et al., 2012). For this study, we focused specifically on the gene sets ‘c2.cp.kegg.v7.5.1.entrez.gmt’ and ’c5.go.bp.v7.5.1.entrez.gmt’ in our GSEA analysis.

Prognosis analysis

Survival data were obtained from the TCGA Pan-Cancer Clinical Data Resource, encompassing metrics such as overall survival, progression-free interval, disease-free interval, and disease-specific survival. This dataset was utilized to assess the prognostic significance of HKDC1. Subsequently, cancer patients were divided into two distinct groups—those with high HKDC1 expression and those with low expression—based on the median HKDC1 levels. To evaluate survival outcomes, we generated Kaplan–Meier survival curves and conducted Cox regression analyses using the R packages ‘survival’ and ‘survminer’, which offer essential tools for performing survival analyses and visualizations in R version 4.2.2 (R Core Team, 2022).

Cell culture

We selected gastric and breast cancer cell lines representing both cancer types for our functional validation studies. Specifically, we utilized the human gastric cancer cell line MKN-1 and the breast cancer cell line MDA-MB-231, both procured from the Shanghai Branch of the Chinese Academy of Sciences (Shanghai, China). MKN-1 cells were cultured in RPMI 1640 medium, whereas MDA-MB-231 cells were grown in Dulbecco’s Modified Eagle Medium supplemented with 10% fetal bovine serum. All cultures were maintained at 37 °C in a humidified incubator with 5% CO2.

Small interfering RNA (siRNA) construction and transfection

The siRNA sequences targeting HKDC1 were synthesized by incorporating them into the pGreen vector. Following an evaluation of various candidate siRNAs, two specific sequences were identified for further experimentation: AGAUGGAGAGUCAGUUCUATT UAGAACUGACUCUCCAUCUTT (designated as si1) and CUUGCUAAUACAAGAGAGATTUCUCUCUUGUAUUAGCAAGTT (designated as si2). Additionally, the negative control is a non-specific control small interfering RNA (siNC). A total of 4 ×105 cells were seeded in each well of 6-well plates and allowed to incubate for 24 h. Following this incubation, the cells were transfected with 10 µl of either HKDC1 si1/2 or siNC per well, using Lipofectamine™ RNAiMAX reagent according to the manufacturer’s instructions. The cells were harvested 48 h post-transfection for evaluation via the CCK-8 assay, as well as for the extraction of RNA and proteins.

RNA extraction and qPCR

We first wait for cells to grow about 60–70% before transfection. Then, when the cell growth was close to 1 × 106 cells, RNA was extracted by using the RNA Quick Purification kit. Primers for qPCR including HKDC1 and β-actin were synthesized by Tianyi Huiyuan (Guangzhou, China). GraphPad Prism was used for statistical analysis. The primer sequence is shown below. HKDC1 forward - GTTGCCCACCTTCGTCAGG, HKDC1 reverse - AGCGACTTGCACCTTCAGC, β-actin forward - CCTGGCACCCAGCACAAT, β-actin reverse-GGGCCGGACTCGTCATAC.

Cell counting kit-8 assay

The assay was performed according to the protocols provided by the manufacturer (GLPBIO). Specifically, the designated cells (1,500 cells per well) were seeded in triplicate in 96-well plates. Cell viability assessments of the specified cells were conducted over a period of six days following the initial seeding. Following this, the culture medium was then removed, and a solution of the cell counting kit-8 (CCK-8) assay, prepared at a dilution of 1:10 (10 µL), was added to each well. Following a 2-hour incubation period, the optical density (OD) of each well was measured at a wavelength of 450 nm.

Migration and invasion assays

Transwell assays were conducted using 24-well transwell plates (Corning Costar) that had been pre-treated with Matrigel (BD Biosciences). Specifically, 15 × 105 transfected cells were added to the upper chamber of the Matrigel-coated transwell chambers. After a 24-hour incubation period, the cells that migrated through the membrane were fixed and stained with a 0.1% crystal violet solution. To assess cell migration, a method similar to that used for the cell invasion assay was implemented; however, in this case, the transwell chambers weren’t coated with 200 mg/ml Matrigel.

Statistical analysis

Statistical analysis was performed with R (version 4.2.2; R Core Team, 2022) and GraphPad Prism 8 software (USA), ns, not significant, *p < 0.05, **p < 0.01, and ***p < 0.001. This study presented all data as means ± standard deviation. Survival analysis was carried out using the Kaplan–Meier approach and log-rank test.

Results

HKDC1 is upregulated in multiple cancer types

The amalgamation of samples from both the GTEx and TCGA databases produced persuasive evidence of a notable escalation in HKDC1 expression across diverse tumor types. This significant upregulation was noted across a diverse array of cancers, encompassing bladder urothelial carcinoma (BLCA), breast cancer (BRCA), bile duct carcinoma (CHOL), stomach adenocarcinoma (STAD), among others (see Fig. 2A). Further validation using a paired Student’s t-test confirmed this upregulation, revealing a statistically significant increase in HKDC1 expression levels in tumor tissues compared to adjacent normal tissues (Fig. 2B). This finding highlights the potential significance of HKDC1 in cancer biology and suggests its prospective role as a therapeutic target or biomarker.

Figure 2 HKDC1 exhibits increased expression across various cancer types.

(A) The examination of HKDC1 mRNA expression across 33 different cancer types was performed using data sourced from the TCGA and GTEx databases. In the analysis, red dots denote tumor (T) tissues, while green dots signify normal (N) tissues. (B) A comparative analysis of HKDC1 mRNA levels in cancer and para-cancer paired samples in TCGA.

HKDC1 is associated with tumor immunity

Utilizing the XCELL algorithm, our results revealed a positive correlation between HKDC1 expression levels and the infiltration of diverse immune cell subsets in numerous cancers (Fig. 3A). This implies that HKDC1 could be instrumental in influencing the immune landscape within the tumor microenvironment.

Figure 3 HKDC1 is associated with the diverse immune cell types across various malignant tumors.

(A) A correlation heatmap illustrates the relationship between HKDC1 expression levels and the infiltration of different immune cell populations, as evaluated using the XCELL algorithm. The transition from red to blue in the heatmap indicates the strength of the correlation coefficient. (B) An additional heatmap displays the expression of HKDC1 with the levels of 11 immune checkpoint genes.

Furthermore, we investigated the relationship between HKDC1 and immune checkpoint molecules, which are vital regulators of immune responses. Our analysis demonstrated a significant correlation between HKDC1 expression and the levels of several key immune checkpoints (Fig. 3B), including PDCD1 (Fig. 3A), BTLA (Fig. 3B), CTLA4 (Fig. 3C), HAVCR2 (Fig. 3D), LAG3 (Fig. 3E), LILRB2 (Fig. 3F), LILRB4 (Fig. 3G), SIGLEC7 (Fig. 3H), SIRPA (Fig. 3I), TIGIT (Fig. 3J), and VSIR (Fig. 3K). These findings underscore the intricate interactions between HKDC1 and the immune checkpoint network, suggesting that HKDC1 may function as a modulator of immune responses and checkpoint signaling, with significant implications for cancer immunotherapy and prognostic outcomes.

HKDC1 serves as an oncogene in multiple cancer types

We utilized GSEA to pinpoint KEGG pathways that are markedly enriched in genes linked to HKDC1. Our analysis, as depicted in Fig. 4A, revealed a robust enrichment of several cancer-relevant pathways, highlighting the multifaceted nature of HKDC1’s involvement in cancer biology. Among the significantly enriched pathways, we observed a notable association with glycolysis (Fig. 4B), indicating that HKDC1 may be involved in modulating cellular energy metabolism, a hallmark of cancer. Additionally, the increased representation of ubiquitin-mediated proteolysis (Fig. 4C) and apoptosis (Fig. 4E) pathways indicates that HKDC1 could be involved in the modulation of protein degradation and the maintenance of cell viability, respectively. The involvement of TCA Cycle (Fig. 4D) and glutathione metabolism (Fig. 4G) underscores the potential impact of HKDC1 on cellular metabolic processes.

Figure 4 The gene HKDC1 is linked to numerous pathways and biological processes pertinent to cancer.

(A) A heatmap displays the KEGG pathways related to HKDC1. (B–M) Lollipop plots further illustrate the biological processes connected to HKDC1, also derived from the GSEA results. Moreover, the color gradient from blue to red indicates the normalized enrichment score (NES) values.

Furthermore, the observed enrichment in the cell cycle pathway (Fig. 4F) and the RNA degradation pathway (Fig. 4L) indicates that HKDC1 may significantly influence cell proliferation and the regulation of gene expression, respectively. Additionally, the participation of glycerophospholipid metabolism (Fig. 4H), lysosome (Fig. 4I), pentose and glucuronate interconversions (Fig. 4J), and the pentose phosphate pathway (Fig. 4K) broadens the range of biological functions attributed to HKDC1, indicating its involvement in membrane dynamics, intracellular degradation processes, and alternative pathways of glucose metabolism.

Lastly, the notable enrichment in the Toll-like receptor signaling pathway (as shown in Fig. 4M) highlights the potential role of HKDC1 in modulating immune responses and inflammatory processes, both of which are increasingly recognized as crucial factors influencing cancer development and therapeutic outcomes. Collectively, our GSEA offers an extensive overview of the prospective functional dynamics associated with HKDC1 across various cancer types, thereby laying the groundwork for subsequent explorations into its distinct roles and underlying mechanisms in diverse oncological contexts.

HKDC1 is regulated by copy number amplification and DNA methylation

We conducted an exhaustive examination of both genetic and epigenetic modifications to investigate the mechanisms underlying the increased expression of HKDC1 in cancer. At the genetic level, we examined mutations and copy number variations (CNVs) within the HKDC1 gene. Our findings revealed a substantial number of mutations distributed uniformly across the entire coding sequence of HKDC1 in diverse cancer types, suggesting a potential role for these mutations in modulating HKDC1 expression. Additionally, we observed a prevalent pattern of HKDC1 copy number deletions across the majority of tumor types, with a minority of samples exhibiting copy number gains (Fig. 5A). These CNVs likely contribute to the deregulation of HKDC1 expression in cancer.

Figure 5 HKDC1 is affected by both copy number amplification and methylation.

(A) A comprehensive analysis of variations in DNA copy number across 33 distinct types of cancer. (B) An evaluation of DNA methylation patterns within these 33 cancer types. Probes highlighted in red indicate their targeting of the promoter region.

In exploring the epigenetic aspects, we analyzed DNA methylation alterations in the promoter region and gene body of HKDC1. Our discoveries indicated that HKDC1 mRNA expression exhibits a negative correlation with methylation levels at multiple sites across the majority of tumor types examined, particularly in adrenocortical carcinoma (ACC), breast carcinoma (BRCA), acute myeloid leukemia (LAML), mesothelioma (MESO), prostate adenocarcinoma (PRAD), testicular germ cell tumors (TGCT), and uveal melanoma (UVM). This inverse correlation suggests that the observed hypomethylation within the HKDC1 promoter and/or gene body may contribute to the elevated expression of HKDC1 in these particular malignancies. Conversely, in certain cancer types, including diffuse large B-cell lymphoma (DLBC), kidney chromophobe (KICH), kidney renal papillary cell carcinoma (KIRP), thyroid carcinoma (THCA), and thymoma (THYM), We observed that the expression of HKDC1 exhibits a positive correlation with methylation levels. This finding indicates that the relationship between methylation and HKDC1 expression may differ according to the specific cancer context, necessitating further investigation. The detailed analysis of both genetic and epigenetic modifications, illustrated in Fig. 5B, offers critical insights into the intricate regulatory mechanisms that influence HKDC1 expression in the context of cancer.

Pan-cancer analysis of the multifaceted prognostic value of HKDC1

To assess the clinical relevance of HKDC1 expression across various cancers, we performed a comprehensive analysis of its association with patient outcomes, utilizing data from 33 cancer types sourced from the TCGA database. Our evaluation focused on several critical metrics, including OS, DSS, DFI and PFI. The findings indicated that HKDC1 transcripts frequently served as negative prognostic indicators for both OS and DSS in multiple cancer types. Notably, higher levels of HKDC1 expression, as evidenced by transcriptomic and gene expression analyses, were consistently linked to poorer OS and DSS outcomes, particularly in malignancies such as thyroid tumor (THCA), pancreatic ductal carcinoma (PAAD), lung squamous cell carcinoma (LUSC), among others (see Figs. 6A and 6B).

Figure 6 The expression levels of HKDC1 are significantly associated with adverse prognostic outcomes.

The results are illustrated in forest plots for (A) OS, (B) DSS, (C) DFI, and (D) PFI, showcasing the relationship between HKDC1 expression and patient outcomes across 33 cancer types as reported in TCGA.

Furthermore, HKDC1 serves as a prognostic indicator influencing DFI and PFI across diverse cancer types (refer to Figs. 6C and 6D). The KM plots further validated the prognostic significance of HKDC1, demonstrating a substantial correlation with adverse outcomes across multiple endpoints and cancer types (see Figs. 6A–6P). Collectively, these findings underscore the clinical importance of HKDC1 expression as a potential biomarker for cancer prognosis and as a promising target for therapeutic intervention.

HKDC1 enhances the growth and movement of gastric and breast cancer cells in an in vitro setting

Acknowledging the pivotal role of HKDC1 in cancer progression, we undertook an in vitro study aimed at elucidating its function within gastric and breast cancer cell lines. To initiate our analysis, we evaluated the protein expression levels of HKDC1 in both MKN-1 and MDA-MB-231 cell lines following siRNA-mediated silencing (Figs. 7A, 7E). At the same time, RNA levels of HKDC1 were also detected (see Figs. 7B, 7F).

Figure 7 The effect of HKDC1 on the growth, invasion, and migration of MKN-1 and MDA-MB-231 cell lines under in vitro conditions was examined.

(A, E) The effectiveness of siRNA-HKDC1 transfection in MKN-1 and MDA-MB-231 cell lines was validated through Western blot analysis. (B, F) Transfection efficiency of qPCR in gastric and breast cancer cells. (C, G) The proliferation rates of these cell lines were evaluated using the Cell Counting Kit-8 assay. (D, H) Additionally, transwell assays were carried out to investigate the migratory and invasive potential of MKN-1 and MDA-MB-231 cells.

To explore the functional implications of HKDC1 downregulation, we conducted a cell counting kit-8 (CCK-8) assay on the previously mentioned HKDC1 knockdown cell lines (refer to Figs. 7C, 7G). The results of this assay revealed that HKDC1 significantly influences the proliferative capacity of both MKN-1 and MDA-MB-231 cells, indicating that HKDC1 plays a positive role in promoting their growth. Furthermore, we performed transwell assays to evaluate the effects of HKDC1 on the migratory and invasive capabilities of MKN-1 and MDA-MB-231 cell lines. As expected, the silencing of HKDC1 significantly reduced the migratory and invasive potential of these cells as they traversed the transwell membrane (see Figs. 7D, 7H).

Discussion

An increasing volume of research suggests that HKDC1 demonstrates distinct expression profiles in tumor tissues and plays crucial roles in multiple facets of tumor biology (Wang et al., 2023). However, to date, the question of whether HKDC1 exerts its influence across different tumor types through shared molecular mechanisms remains unanswered. Despite extensive literature searches, no comprehensive pan-cancer analysis of HKDC1 from a holistic viewpoint has been reported, underscoring the need for a global investigation into its role in tumorigenesis. In order to fill this existing knowledge gap, we undertook a comprehensive examination of HKDC1 across 33 different types of tumors, utilizing the vast dataset provided by TCGA. This methodology enabled us to investigate not only the expression profiles of HKDC1 but also its relationships with significant molecular features such as survival outcomes, genetic modifications, and immune cell infiltration. By embracing this holistic viewpoint, our goal was to achieve an enhanced comprehension of the pervasive and possibly integrative function of HKDC1 in cancer, thereby aiding in the formulation of more focused and efficacious therapeutic approaches.

Our study’s findings highlight a complex expression profile of HKDC1 across various cancer types, with significantly elevated levels observed in the malignant tissues of cholangiocarcinoma (CHOL), pancreatic adenocarcinoma (PAAD), glioblastoma multiforme (GBM), liver hepatocellular carcinoma (LIHC), ovarian cancer (OV), rectal adenocarcinoma (READ), stomach adenocarcinoma (STAD), and uterine corpus endometrial carcinoma (UCEC) compared to their respective healthy tissues. This observation aligns with previous research focusing on hepatocellular carcinoma, further validating the potential significance of HKDC1 in tumor biology. Conversely, we noted decreased HKDC1 expression in KICH, LUSC, PRAD, and THCA, suggesting that distinct cancer types may harbor varied underlying mechanisms that govern HKDC1 expression levels. These discrepancies underscore the importance of elucidating the specific roles and processes underpinning HKDC1 regulation in individual cancer types.

In light of these observations, HKDC1 has emerged as a potential biomarker for cancer prognosis. To evaluate its prognostic significance, we conducted an analysis examining its association with tumor-specific outcomes, including overall survival (OS), disease-specific survival (DSS), disease-free interval (DFI), and progression-free interval (PFI). Nevertheless, our findings revealed a degree of uncertainty regarding the exact prognostic function of HKDC1, as it exhibited an association with unfavorable prognosis in certain tumor types while serving as a protective factor in others, specifically in THYM and LGG. These divergent outcomes highlight the intricate role of HKDC1 in cancer prognosis and underline the necessity for additional investigations to elucidate its context-dependent effects across various cancer types.

Extensive research has established that tumor cells participate in complex interactions and co-evolutionary dynamics with their surrounding microenvironment (Loh & Ma, 2024), which consequently aids in the progression of tumors (Mempel, Lill & Altenburger, 2024). Simultaneously, there is an increasing clinical focus on the creation of effective immunotherapeutic strategies for cancer patients, leading to the emergence of numerous promising treatment options (Mukherjee, Biswas & Roy, 2024). In this study, we identified a significant correlation between HKDC1 expression levels and the infiltration of various immune cell populations within the tumor microenvironment across multiple cancer types (Zhang et al., 2024). This observation highlights the potential role of HKDC1 in shaping the immune landscape of tumors.

Furthermore, our study has established a significant relationship between HKDC1 expression levels and the upregulation of various immune checkpoints and immunomodulatory molecules. Given the crucial role of immune checkpoints in enabling immune evasion, these findings suggest that HKDC1 may be integral to the interplay between tumor cells and the immune system (Kroemer et al., 2024). These molecules have emerged as central targets in pharmaceutical research, with accumulating evidence suggesting that inhibiting immune checkpoints could represent a promising strategy in the field of cancer immunotherapy (Bandini, Ulivi & Rossi, 2024; Dong et al., 2022). Consequently, the association identified between HKDC1 and immune checkpoint expression in this research further emphasizes the potential importance of HKDC1 within the landscape of cancer immunotherapy. Some studies revealed the specific mechanism of hexokinase domain component 1 (HKDC1) and immune checkpoint PD-L1 expression in hepatocellular carcinoma (HCC). HKDC1 promotes the phosphorylation and nuclear translocation of STAT1 by binding to cytoskeletal protein ACTA2, thereby enhancing the expression of PD-L1. The study further demonstrated that the combination of inhibiting HKDC1 and anti-PD-1 /PD-L1 antibodies can significantly enhance the anti-tumor response of CD8+ T cells. This combination treatment strategy showed significant anti-tumor effects in a mouse model of HCC and extended survival in the mice. In the future, the combined treatment strategy of inhibiting HKDC1 and anti-PD-1 /PD-L1 antibodies has shown significant anti-tumor effects, and may become an important direction of tumor immunotherapy in the future (Zhang et al., 2024).

Moreover, our findings reveal that, in most of the cancer types analyzed, HKDC1 expression is positively correlated with tumor stemness—an aspect previously associated with the promotion of tumor initiation, metastasis, and treatment resistance (Zhang et al., 2016). In addition, Fan et al. (2024) revealed the function of HKDC1 under hypoxic conditions, suggesting that HKDC1 can respond to changes in tumor microenvironment and maintain the proliferation ability of liver cancer cells. HKDC1 promotes the stem cell properties of cancer cells by binding to glycogen synthase kinase 3β (GSK3β), which stabilizes beta-catenin. Overall, this study provides a mechanistic basis for the application of HKDC1 as a potential therapeutic target in advanced liver cancer, and provides new ideas for the treatment of tumor patients (Yu & Bao, 2023). This finding highlights a potential connection between HKDC1 and tumor progression, as well as its possible impact on the effectiveness of therapeutic strategies.

From both functional and mechanistic viewpoints, GSEA has indicated that HKDC1 may be crucial in several essential cancer hallmark pathways and their related biological processes. Notably, research has demonstrated that HKDC1 significantly influences cell cycle regulation and DNA replication mechanisms. A recent study (Guo etal., 2022) emphasized that cancer cells exploit specific post-translational modifications of HKDC1 to inhibit its function during apoptosis, thereby facilitating the survival of malignant cells. Additionally, there is evidence that the increased expression of HKDC1 correlates with the advancement of tumor angiogenesis, along with heightened cancer invasion and metastasis (Zhao et al., 2023). In terms of clinical application, our GSEA analysis found that HKDC1 is associated with apoptosis pathways in a variety of tumors. Previous studies have found that the hexokinase family binds to the voltage-dependent anion channel (VDAC) in the outer membrane of mitochondria, thereby promoting the glycolysis of tumor cells and inhibiting the apoptosis of tumor cells. In theory, HKDC1, a newly discovered member of hexokinase, should also have the ability to bind. Therefore, in tumor treatment, compounds can be used to promote the dissociation of hexokinase from VDAC to promote apoptosis, so as to treat tumors. At present, the apoptosis of breast cancer cells MDA-MB-231 can be triggered by flavonoid FV-429. At the same time, pharmacological studies have confirmed that inhibitors of hexokinase (2-DG) inhibit the growth of lung cancer cells in humans and mice by inducing apoptosis and autophagy, and autophagy is also believed to play an important role in chemical drug resistance of cancer cells. For example, 3-hydroxypyruvate, a hexokinase inhibitor, induces tumor autophagy in MDA-MB-231 cells of breast cancer. Inhibition of autophagy may be an innovative strategy for adjuvant chemotherapy in breast cancer (Zhou et al., 2016).

In the present study, we performed in vitro experiments aimed at further elucidating the stimulatory role of HKDC1 in cancer-related processes. Our results demonstrate that the increased expression of HKDC1 markedly enhances cell proliferation and migration. Mechanistically, our comprehensive multi-omics analysis has shown that the elevated levels of HKDC1 are affected by both amplification of the DNA copy number and changes in promoter methylation. These findings offer essential insights into the intricate regulatory mechanisms that govern the function of HKDC1 in cancer progression.

In summary, the results of this investigation provide valuable insights into the diverse roles of HKDC1 in tumor regulation, highlighting several critical aspects. To begin with, HKDC1 exhibits a pivotal function in tumor biology, akin to GAPDH, which is frequently observed to be overexpressed across various cancer types. Moreover, the heightened expression of HKDC1 is profoundly affected by changes in DNA copy number and epigenetic modifications, particularly promoter methylation. Additionally, the expression levels of HKDC1 hold substantial clinical importance for individuals diagnosed with tumors, serving as a prognostic biomarker that may inform therapeutic strategies and affect patient prognoses. These findings highlight the urgent need for further exploration into the mechanisms that regulate HKDC1’s involvement in tumor dynamics and its prospective application as a biomarker in cancer therapies.

Supplemental Information

Supplemental Information 1 DKDC1 and immune checkpoints are shown in detail

(A) PDCD1, (B) BTLA, (C) CTLA4, (D) HAVCR2, (E) LAG3, (F) LILRB2, (G) LILRB4, (H) SIGLEC7, (I) SIRPA, (J) TIGIT, and (K) VSIR across various cancer types.

Supplemental Information 2 Kaplan-Meier survival curves were used to characterize the correlation between HKDC1 and prognosis

Additionally, Kaplan-Meier survival curves have been generated to highlight the correlations between HKDC1 mRNA expression and OS, DSS, DFI, and PFI among patients, as depicted in panels (A–P).

Supplemental Information 3 Western blot for BRCA and STAD

Supplemental Information 4 STAD knockout efficiency qPCR results

Gastric cancer knockout efficiency qPCR results .

Supplemental Information 5 BRCA knockout efficiency qPCR results

Breast cancer knockout efficiency qPCR results.

We appreciated the General Surgery Laboratory of the First Affiliated Hospital of Sun Yat-sen University for providing the experimental platform and facilities.

Abbreviations

ACC Adrenocortical carcinoma

BLCA Bladder Urothelial Carcinoma

BRCA Breast invasive carcinoma

HNSC Head and Neck squamous cell carcinoma

CHOL Cholangiocarcinoma

GBM Glioblastoma multiforme

ESCA Esophagus carcinoma

KICH Kidney Chromophobe

LAML Acute Myeloid Leukemia

LGG Brain Lower Grade Glioma

COAD Colon adenocarcinoma

READ Rectum adenocarcinoma

READ Rectum adenocarcinoma Esophageal carcinoma

TGCT Testicular Germ Cell Tumors

THCA Thyroid carcinoma

SKCM Skin Cutaneous Melanoma

KIRC Kidney renal clear cell carcinoma

TCGA The Cancer Genome Atlas

KIRP Kidney renal papillary cell carcinoma

THYM Thymoma

LIHC Liver hepatocellular carcinoma

SARC Sarcoma

LUAD Lung adenocarcinoma

STAD Stomach adenocarcinoma

LUSC Lung squamous cell carcinoma

OS Overall survival

MESO Mesothelioma

PFI Progression-free interval

OV Ovarian serous cystadenocarcinoma

DSS Disease-Specific Survival

PAAD Pancreatic adenocarcinoma

DFI Disease-Free Interval

PRAD Prostate adenocarcinoma

CESC Cervical squamous cell carcinoma and endocervical adenocarcinoma

DLBC Lymphoid Neoplasm Difuse Large B-cell Lymphoma

Additional Information and Declarations

Competing Interests

Author Contributions

Data Availability

The authors declare there are no competing interests.

Zhi Liang conceived and designed the experiments, performed the experiments, prepared figures and/or tables, and approved the final draft.

Tianhao Zhang performed the experiments, prepared figures and/or tables, and approved the final draft.

Jiajia Huang analyzed the data, prepared figures and/or tables, and approved the final draft.

Zhixin Huang analyzed the data, prepared figures and/or tables, and approved the final draft.

Zeyu Zhao analyzed the data, prepared figures and/or tables, and approved the final draft.

Shirong Cai analyzed the data, authored or reviewed drafts of the article, and approved the final draft.

Jinping Ma analyzed the data, authored or reviewed drafts of the article, and approved the final draft.

The following information was supplied regarding data availability:

The data is available at figshare:

– liang, zhi (2025). BRCA .zip. figshare. Dataset. https://doi.org/10.6084/m9.figshare.27193869.v1

– liang, zhi (2025). STAD.zip. figshare. Dataset. https://doi.org/10.6084/m9.figshare.27193743.v1.

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
