# Peer review of "A comprehensive prognostic and immunological analysis of hexokinase domain containing protein-1 (HKDC1) in pan-cancer"

_PeerJ, doi:10.7717/peerj.19083_

## Round 0.1 · original submission · Major Revisions

Three reviewers found the importance of your research. However, they also raise lots of critical comments. I agree with their comments. In particular, it is important that the sample sizes (N values) for the cancers analyzed are consistent. Please make revisions according to their comments.

·

Basic reporting

1. There are numerous grammatical errors throughout. Typos such as "cancer" are common.
a. Inconsistent use of professional language and minor grammatical errors throughout the manuscript (e.g., "tumor dryness" should be clarified or replaced with precise terminology).
b. Awkward phrasing in some sections reduces clarity (e.g., "HKDC1 expression is positively correlated with tumor dryness").
c. Some sentences are overly complex, making comprehension difficult for readers unfamiliar with the subject.
2. The manuscript does add new knowledge to the field as a whole, but in its present form, it's not useful. Anyone who knows how to use cbioportal or UCSC genome browser can get this data.
3. What is interesting is the section on immunology and how HKDC1 can be regulated. However, these are mere figures with little to no explanation given in the results or discussion sections.
If these data are explained, this can be a wealth of knowledge. For example Fig 5 is just colors and numbers - no idea what any of these mean.
4. Provide additional background on HKDC1 to strengthen the context.
5. The manuscript should explicitly address ethical considerations related to data use from TCGA and GTEx databases.
6. The statistical rigor of subgroup analyses could be improved, and methods should be explicitly detailed to prevent misinterpretation.

Experimental design

Strengths: The methods are described with sufficient detail to allow replication. However, more information on the selection criteria for the cancer types analyzed could be beneficial. The use of TCGA and GTEx databases is appropriate, and the integration of these datasets is well-executed.

1. Limited detail in certain methodological sections (e.g., exact criteria for "high" and "low" HKDC1 expression groups).
2. No clear discussion on potential biases introduced by combining TCGA and GTEx data despite eliminating batch effects.
3. Insufficient detail on statistical methods used in correlation analyses, especially for immune cell infiltration and immune checkpoints.

Validity of the findings

1. Overgeneralization of findings. For example, HKDC1 is linked to both pro- and anti-cancer effects depending on the cancer type, but these distinctions are not explored in depth.
2. Limited exploration of context-specific roles of HKDC1 across different cancer types.
3. The claim of HKDC1 being a "promising therapeutic target" lacks supporting preclinical or translational evidence.
4. Repetition of points made in the results without sufficient critical analysis.
5. Limited integration of the findings into the broader field of cancer biology and metabolism.
6. Missed opportunities to discuss limitations, such as potential confounders in TCGA data or the lack of in vivo validation.
7. Conclusions are overly broad and not sufficiently nuanced to align with the variability of HKDC1's roles across cancer types.
8. The statement about HKDC1 as a "prognostic biomarker" needs stronger evidence and clearer delineation of cancer-specific prognostic implications.

Additional comments

1. Figures lack sufficient detail in captions, such as descriptions of axes or methodologies used (e.g., Figure 4 heatmaps and lollipop plots).
2. The resolution of some images is suboptimal, which may hinder detailed examination.
3. Figure labels (e.g., A, B, C) could be clearer and more consistently aligned with textual references.

·

Basic reporting

This article utilizes databases such as TCGA to investigate the expression, immunological, and prognostic relevance of HKDC1 (Hexokinase Domain-Containing Protein 1) in various cancer types. It highlights HKDC1's oncogenic role in multiple cancers and its potential as a significant prognostic biomarker.

Below are the comments on the article:
Introduction Section:
The article should focus more on tumor metabolism and the current state of research on HKDC1 to provide readers with a clearer understanding of the background related to HKDC1. Additionally, a search of journals from sources like PubMed shows that there is ongoing research on HKDC1 in many cancers. However, the article does not sufficiently address HKDC1's specific roles in cancer progression or metabolism. Although the introduction mentions a comprehensive analysis of HKDC1, the content primarily focuses on prognostic analysis and immune relevance. Therefore, this section should emphasize these aspects more explicitly. Overall, the introduction should prioritize cancer metabolism and the importance of the HK family, the existing research on HKDC1, its limitations, and the study's objectives and methods.

Experimental design

no comment

Validity of the findings

Results Section:
Regarding Figures 2A and 2B, it is unclear whether the sample sizes (N values) are consistent. Additionally, in cancer types such as BLCA, BRCA, PAAD, PRAD, READ, and UCEC, different analytical methods yield varying results, which the authors should address and discuss.
For the findings presented in Figure 3, the authors discuss the correlation between HKDC1 and tumor immunity, highlighting its potential role in modulating the tumor microenvironment, particularly through its relevance to immune checkpoint genes. However, the visualization in Figure 3A makes it difficult to establish a direct connection between HKDC1 and tumor immunity. While the authors state that HKDC1 positively correlates with immune cell infiltration, certain cases, such as PAAD and CESC, show negative correlations. Similarly, Figure 3B fails to provide a cohesive interpretation that supports a definitive positive correlation between HKDC1 and immune checkpoints.

Additional comments

The discussion primarily reiterates the results. While relevant literature is cited, the authors fail to delve deeply into how this study differs from or improves upon existing research, thus failing to underscore the study's innovation. Moreover, the discussion focuses on supportive results and lacks exploration of cases where HKDC1 expression is low or where its prognostic relevance is insignificant in certain cancers. It is recommended to incorporate examples of how HKDC1's role could be further validated in immune therapy and cancer metabolism-targeted treatments.

Overall Evaluation:
Although the study focuses on the expression of HKDC1 across cancers and its role in immune regulation, the lack of effective synthesis and summary hinders the clarity of the article.
Discussion Section

Reviewer 3 ·

Basic reporting

Pass. I believe the manuscript is well-written and requires only minor modifications in the discussion section.

Experimental design

Pass. I believe the manuscript is acceptable as the experimental design is well-constructed and reasonable.

Validity of the findings

Fail. I believe some data needs to be revalidated before the manuscript can be considered for publication.

Additional comments

Comments:
Due to the critical role of metabolic switching in a glucose-independent manner in most malignant cancer cells, the author aims to elucidate the significance of hexokinase domain-containing protein 1 (HKDC1) across various cancer types, focusing on multiple aspects such as immunity and oncogenesis. The data reveal that copy number variations (CNVs) and DNA methylation are closely associated with the regulation of HKDC1 in cancers. Notably, HKDC1 exhibits multifaceted prognostic potential. In vitro assays further emphasize the pivotal role of HKDC1 in regulating the growth and invasive abilities of gastric and breast cancers. Utilizing the extensive dataset provided by The Cancer Genome Atlas (TCGA), the authors demonstrate that upregulated HKDC1 plays a key role in the regulation of metabolism, immunity, growth, and invasion in malignant cancers. While this manuscript holds significant potential for publication, addressing several minor concerns could greatly enhance its quality.

Major Revisions:
1. In Figure 7, the Western blot data indicate that the knockdown efficiency of HKDC1 is suboptimal. Please confirm whether the siRNA oligos are effectively targeting HKDC1. If possible, validate the knockdown efficiency and repeat the critical experiments to ensure the accuracy and reliability of the data.
2. In Figure 3 (C-M), please clearly specify the names of the cancers being analyzed to enhance clarity and understanding.
3. In Figure 6 (G, M, Q, T) and (J, P, R), there appear to be repetitive cancer names. Please confirm if this is accurate, and if not, kindly make the necessary corrections.
4. Please cite recent references regarding HKDC1's role in immunity regulation and drug resistance in the discussion section.

Annotated reviews are not available for download in order to protect the identity of reviewers who chose to remain anonymous.

---

## Round 0.2 · accepted · Accept

The reviewers confirm that the authors have addressed all of the reviewers' comments. Your manuscript is now ready for publication.

·

Basic reporting

Some typos and grammatical mistakes still exist.

Experimental design

No comment

Validity of the findings

No comment

Additional comments

Resolution of the images is unchanged.

Reviewer 3 ·

Basic reporting

no comment

Experimental design

no comment

Validity of the findings

no comment

Additional comments

Thank you for responding to my feedback point by point; all the answers are satisfactory.